# Vancomycin and Linezolid-Resistant Enterococcus Isolates from a Tertiary Care Center in India

**DOI:** 10.3390/diagnostics13050945

**Published:** 2023-03-02

**Authors:** Mallika Sengupta, Riya Sarkar, Soma Sarkar, Manideepa Sengupta, Sougata Ghosh, Parthajit Banerjee

**Affiliations:** 1Department of Microbiology, All India Institute of Medical Sciences (AIIMS), Kalyani 741245, India; 2Vijaya Diagnostics Laboratory, Hyderabad 500029, India; 3Department of Microbiology, NRS Medical College, Kolkata 700014, India; 4Department of Microbiology, Medical College, Kolkata 700073, India; 5Department of Microbiology, KPC Medical College, Kolkata 700032, India

**Keywords:** VRE, LRE, resistant, Enterococcus

## Abstract

Introduction: There is increasing development of antibiotic resistance among the Enterococcus species. Objectives: This study was performed to determine prevalence and characterize the vancomycin-resistant and linezolid-resistant enterococcus isolates from a tertiary care center. Moreover, the antimicrobial susceptibility pattern of these isolates was also determined. Materials and Methods: A prospective study was performed in Medical College, Kolkata, India, over a period of two years (from January 2018 to December 2019). After obtaining clearance from the Institutional Ethics Committee, Enterococcus isolates from various samples were included in the present investigation. In addition to the various conventional biochemical tests, the VITEK 2 Compact system was used to identify the Enterococcus species. The isolates were tested for antimicrobial susceptibility to different antibiotics using the Kirby–Bauer disk diffusion method and VITEK 2 Compact to determine the minimum inhibitory concentration (MIC). The Clinical and Laboratory Standards Institute (CLSI) 2017 guidelines were used to interpret susceptibility. Multiplex PCR was performed for genetic characterization of the vancomycin-resistant Enterococcus isolates and sequencing was performed for characterization of the linezolid-resistant Enterococcus isolates. Results: During the period of two years, 371 isolates of *Enterococcus* spp. were obtained from 4934 clinical isolates showing a prevalence of 7.52%. Among these isolates, 239 (64.42%) were *Enterococcus faecalis*, 114 (30.72%) *Enterococcus faecium*, and others were *Enterococcus durans*, *Enterococcus casseliflavus*, *Enterococcus gallinarum*, and *Enterococcus avium*. Among these, 24 (6.47%) were VRE (Vancomycin-Resistant Enterococcus) of which 18 isolates were Van A type and six isolates of *Enterococcus casseliflavus* and *Enterococcus gallinarum* were resistant VanC type. There were two linezolid-resistant Enterococcus, and they were found to have the G2576T mutation. Among the 371 isolates, 252 (67.92%) were multi-drug resistant. Conclusion: This study found an increasing prevalence of vancomycin-resistant Enterococcus isolates. There is also an alarming prevalence of multidrug resistance among these isolates.

## 1. Introduction

Enterococcus is a genus of facultatively anaerobic, Gram-positive organisms of ovoid shape found in pairs or short chains. Previously, they were classified as Streptococcus Group D [1]. Nosocomial infections are often caused by Enterococci, which are known as opportunistic pathogens. *Enterococcus faecalis* and *Enterococcus faecium* are two of the most common Enterococcus species that are associated with human diseases. Infections caused by them include bacteremia, endocarditis, urinary tract infections, surgical wound infections, and intra-abdominal and intra-pelvic infections. Vancomycin-resistant Enterococci have been on the rise over the last two decades [2]. There has been an increase in resistance to the most common anti-Enterococcal antibiotics, including ampicillin and aminoglycosides, and they are inherently resistant to many other antibiotics, such as cephalosporins and clindamycin, making these infections difficult to treat [3]. 

Infections caused by Enterococcus can be treated with glycopeptide antibiotics. However, glycopeptide resistance is also on the rise. There are six types of glycopeptide resistance described in Enterococci, based on the sequence of the structural gene for the resistance ligase (vanA, vanB, vanC, vanD, vanE, and vanG). The VanA type of resistance is characterized by a high level of resistance to vancomycin and teicoplanin. In contrast, the VanB type is characterized by variable levels of resistance to vancomycin and teicoplanin. VanD strains are resistant to moderate levels of vancomycin and teicoplanin. VanC, VanE, and VanG isolates exhibit low-level resistance to vancomycin only [4]. Vancomycin resistance was found in 24% of Enterococcus isolates in a study by Phukan et al. [5]. The combination of penicillin and gentamicin is no longer effective due to high-level resistance to aminoglycosides. A study performed in Assam, India, found that 53.76% of patients had a high level of gentamicin resistance and 33.33% had a high level of streptomycin resistance [6]. The first report of a linezolid-resistant Enterococcus in India came from Kolkata, but there have been very few reports since then. A G2576T mutation in domain V of 23S ribosomal ribonucleic acid (rRNA) genes of Enterococcus causes clinical resistance to linezolid [7].

Both intrinsic and acquired resistance to many antimicrobials is known to exist in Enterococcus species. There are many resistance genes present that act against various antimicrobials, and this is the most common mechanism responsible for intrinsic resistance. The acquired resistance among Enterococci is caused by DNA mutation or by acquiring new genes through gene transfer. The result is the development of resistance to a variety of antibiotics, including vancomycin, tetracycline, macrolides, fluoroquinolones, etc. Multidrug-resistant isolates are those that are resistant to three or more antimicrobial classes [8]. There has been an increase in multidrug-resistant bacteria (MDR) in clinical and environmental specimens over the last 50 years. Multidrug-resistant organisms are also known as superbugs. Among the most dreaded multidrug-resistant organisms are Gram negative bacilli such as *Klebsiella pneumoniae*, *Acinetobacter baumannii*, *Pseudomonas aeruginosa*, and *Enterobacter* spp. In contrast, Gram-positive bacteria such as *Staphylococcus aureus* and *Enterococcus faecium* have also been reported to display multidrug resistance [9]. To develop resistance to antimicrobials, bacteria have developed a variety of mechanisms. Resistance is caused by several mechanisms. The most significant of these is horizontal gene transfer. Biofilms are also produced by some bacteria. Biofilms remain adherent to the surface and help the bacteria to evade the attack of different antimicrobials [10]. 

Infections due to vancomycin-resistant Enterococci (VRE) have gained prominence as the leading cause of healthcare-associated infections. It is essential to understand the epidemiology of VRE infections, transmission modes in health care settings, and risk factors for colonization and infection. This is essential to prevent and control VRE infections. A hospital’s infection control strategy should be tailored to meet the needs of patients as well as the available resources to effectively manage VRE infections. To decrease the risk of VRE acquisition, it is essential to maintain proper hand hygiene. Prevention and control of VREs include cleaning environment, bathing with chlorhexidine, and adhering to antimicrobial stewardship [11]. There are studies showing that a chlorohexidine gluconate bathing regimen may reduce the incidence of vancomycin-resistant Enterococci (VRE) and methicillin-resistant *Staphylococcus aureus* (MRSA) hospital-acquired infections [12]. 

The objective of this study was to determine the prevalence of Enterococcus isolates and characterize those that were resistant to vancomycin and linezolid in a tertiary health care facility. A pattern of antimicrobial susceptibility was also studied in these cases.

## 2. Materials and Methods

Sample selection: A prospective study was performed in Medical College, Kolkata, India over a period of two years (from January 2018 to December 2019). After obtaining clearance from the Institutional Ethics Committee, non-repetitive clinical samples, such as blood, cerebrospinal fluid (CSF), pus, tissue, urine etc. from locations where isolates of Enterococcus have clinical significance, were included in the study. The clinical samples were received, and a direct Gram stain and culture was made using standard microbiological techniques. A semi-quantitative urine culture was made for all samples as per standard criteria. After the culture was made, a colony count was performed wherever applicable, and samples with significant growth of Enterococcus were included in the study. Enterococcus genus identification was made by Gram stain, non-fastidious growth, and conventional biochemical tests including catalase test, growth on 6.5% NaCl, MacConkey agar, bile esculin agar, and arginine hydrolysis. Fermentation of mannitol, arabinose, sorbitol, growth on tellurite agar and identification by VITEK 2 Compact (BioMerieux Inc., France) was used to identify species. 

Antimicrobial susceptibility testing: These isolates were tested for antimicrobial susceptibility to different antibiotics such as ampicillin (10 µg), tetracycline (30 µg), ciprofloxacin (5 µg), levofloxacin (5 µg), vancomycin (30 µg), teicoplanin (30 µg), and linezolid (30 µg) for all isolates, fosfomycin (200 µg) and nitrofurantoin (300 µg) for urinary isolates, and erythromycin (15 µg) and chloramphenicol (30 µg) for non-urinary isolates. The Kirby–Bauer disk diffusion method was used to test the antimicrobial susceptibility of bacteria on Mueller Hinton agar plates using standard microbiological techniques as per Clinical and Laboratory Standards Institute (CLSI) guidelines. VITEK 2 Compact (BioMerieux Inc., France) was used to determine minimum inhibitory concentrations (MICs) for penicillin, tetracycline, ciprofloxacin, levofloxacin, vancomycin, and teicoplanin, linezolid, and nitrofurantoin. An interpretation of susceptibility was performed according to the CLSI 2017 guidelines [13]. Disk diffusion for fosfomycin was carried out on Mueller Hinton agar supplemented with 25 µg/mL G6P with 200 µg disks. The minimum inhibitory concentration (MIC) for vancomycin, teicoplanin and linezolid was assessed by microbroth dilution method. The MIC for fosfomycin was performed by the agar dilution method. All interpretations of susceptibility pattern were assessed according to the Clinical and Laboratory Standards Institute (CLSI) guidelines 2017. The susceptibility to high-level gentamicin (120 µg) was tested using the Kirby–Bauer disc diffusion method and interpreted using the 2016 European Committee on Antimicrobial Susceptibility Testing (EUCAST) guidelines. Antimicrobial susceptibility testing was conducted with *Staphylococcus aureus* ATCC 25923 and *Enterococcus faecalis* ATCC 29212 for quality control.

Genotypic characterization: The isolates resistant to vancomycin were taken for DNA isolation and amplification. A few colonies were picked from a freshly streaked blood agar plate and inoculated in a 3 mL nutrient broth where they were grown at 37 °C for 2–3 h. The DNA extraction was performed using a Qiagen kit (DNeasy PowerLyzer Microbial Kit, Qiagen, Germany). Multiplex PCR was carried out to detect the presence of genes encoding for vancomycin resistance in the Enterococcus isolates as per the protocol given by Bhatt et al. It was attempted to identify the most common vancomycin-resistant genotypes in Enterococci, namely vanA, vanB, and vanC (vanC1 or vanC2/C3 genes), vanD, vanE, and vanG. For amplification, the following thermal cycling profile was used: 3 min at 94 °C for denaturation, 1 min at 94 °C, 1 min at 45 °C, and 1 min at 72 °C, followed by 7 min at 72 °C for extension. Analyses of DNA fragments were conducted by electrophoresis in 0.5× Tris-borate-EDTA on a 1% agarose gel stained with ethidium bromide [14]. The genomic DNA that was isolated was used for 23S rRNA sequencing. The sequencing was performed on an Illumina sequencer (Lifecell sequencing services Pvt Ltd., India) and the data were obtained. G2576T mutation was noted as a marker for linezolid resistance. 

Data analysis: All data were entered in the excel spreadsheet (Microsoft Office, Redmond, WA, USA). The geometric mean (GM) and the standard deviation (SD) were calculated using excel spreadsheet. The statistical analysis of the data was performed using Statistical Package for Social Sciences (SPSS) version 23 (IBM, SPSS Inc., USA). The data were summarized using mean along with standard deviation for continuous variables, and frequency along with percentages for categorical variable. The Chi square test was used to check the categorical variables association and *p* value < 0.05 was taken as significant.

## 3. Results

During a period of two years, 371 isolates of *Enterococcus* spp. were obtained from 4934 clinical isolates showing a prevalence of 7.52%. Among these 371 samples, there were 208 (56%) males and 163 (44%) females. Among these 371 isolates, 58 (15.63%) were over 18 years, 94 (25.33%) were aged 18–40 years, 132 (35.58%) were aged 41-64 years, and 87 (23.45%) were over 65 years of age.

The isolated Enterococcus samples were obtained from different departments. Most of the isolates were obtained from the Medicine Department (144 (38.81%)), followed by Surgery (112 (30.19%)), Pediatrics (56 (15.09%)), Orthopedics (53 (14.28%)) and Obstetrics and Gynecology (6 (1.62%)).

Out of these 371 isolates, the most common sample was urine (223 (60.11%)), which made up more than half of the samples, followed by blood (76 (20.48%)), pus (68 (18.33%)), and tissue, as shown in Table 1.

Among these 371 isolates of *Enterococcus* spp., 239 (64.42%) were *Enterococcus faecalis*, 114 (30.72%) *Enterococcus faecium* and the others were *Enterococcus durans*, *Enterococcus casseliflavus*, *Enterococcus gallinarum*, and *Enterococcus avium*, as shown in Table 2.

Among these, 24 (6.47%) were VRE (Vancomycin-resistant Enterococcus) of which 18 isolates were Van A type consisting of 14 (5.86%) isolates of *Enterococcus faecalis* and 4 (3.51%) isolates of *Enterococcus faecium.* The other six isolates of VRE consisted of four *Enterococcus casseliflavus* and two *Enterococcus gallinarum* which were intrinsically resistant and known to be of VanC type as given in Table 2.

Among these 371 isolates, Enterococcus species were highly susceptible to vancomycin, teicoplanin, and linezolid. The isolates from urine samples were also highly susceptible to fosfomycin and nitrofurantoin. The susceptibility pattern of these Enterococcus isolates is shown in Table 3. 

Among the 371 isolates, 252 (67.92%) were multi-drug-resistant, i.e., resistant to three different classes of antibiotics. There were two linezolid-resistant Enterococcus, and it was found to have the G2576T mutation in 23S rRNA.

The susceptibility of *E. faecalis* and *E. faecium* was compared, and there was a significant difference in the susceptibility toward ampicillin and nitrofurantoin only, as shown in Table 4.

## 4. Discussion

According to a study performed by Chakraborty et al. in Kolkata in 2011, there was a prevalence of 7.3% Enterococcus isolates from all clinical samples [15]. According to this study, there is a prevalence of 7.52% among all samples, which is very similar to that found in the previous study. In a similar study performed by Phukan et al., the prevalence of Enterococcus isolates was found to be 7.4% [5].

The Vitek 2 automated system was found to be as accurate in speciating 93 Enterococcus species as conventional biochemical tests performed in Assam. *E. faecalis* was the most common isolated species (81.72%), followed by *E. faecium* (12.9%), *E. raffinosus* (3.23%, *n* = 3), *E. avium* (1.08%, *n* = 1) and *E. gallinarum* (1.08%, *n* = 1) [6]. In this study, among these 371 isolates there were 239 (64.42%) *Enterococcus faecalis*, 114 (30.72%) *Enterococcus faecium*, 4 (1.08%) *Enterococcus casseliflavus*, 2 (0.54%) *Enterococcus gallinarum*, 4 (1.08%) *Enterococcus durans* and 8 (2.16%) *Enterococcus avium*. Another study in Uttar Pradesh found that out of the included Enterococcus strains, 47 were *E. faecalis*, 51 were *E. faecium*, two were *E. gallinarum* and one was *E. casseliflavus* [16]. 

Glycopeptide resistance is associated with a variety of phenotypes in Enterococci. *Enterococcus gallinarum* and *Enterococcus casseliflavus*/*flavescens* exhibit intrinsic low-level vancomycin resistance. The VanC-1 ligase is specific for *E. gallinarum*, and the VanC-2/3 ligase is specific for *E. casseliflavus/flavescens*. During pentapeptide peptidoglycan synthesis, VanC enzymes are involved in the formation of D-alanyl-D-serine peptidoglycan precursors that have reduced affinity for vancomycin. Teicoplanin remains effective against organisms resistant to VanC. In these species, vancomycin resistance is a naturally occurring trait, and the associated genes are chromosomal encoded [17]. This study also found that *E.gallinarum* and *E.casseliflavus* isolates were resistant to vancomycin but susceptible to teicoplanin. *Enterococcus gallinarum* and *Enterococcus casseliflavus/flavescens* are Enterococci intrinsically resistant to vancomycin that belong to the *E. gallinarum* group. They are responsible mainly for healthcare-associated infections, such as those associated with blood, urinary tract, and surgical wounds. Globally, these bacteria are causing a significant increase in diseases because they are prone to causing infection in patients with concurrent hepatobiliary or onco-hematologic conditions. Furthermore, their intrinsic vancomycin resistance poses a different infection control problem from that of *Enterococcus faecalis* and *Enterococcus faecium*, which are spread by transmissible plasmids [18].

The prevalence of multidrug-resistant Enterococcus in this study was 67.92%. Similarly, Bhatt et al. found a prevalence of multidrug resistance of 63% among 200 clinical isolates of Enterococcus [19]. According to Praharaj and colleagues, out of 367 isolates of Enterococcus, 32 (8.7%) were vancomycin-resistant. There was no resistance to linezolid among the Enterococcus isolates [20]. Vancomycin-resistant Enterococci were isolated in 24 (6.47%) of the isolates in this study. There were more vancomycin-resistant isolates in Mangalore, where 13 (8.6%) of 150 isolates tested showed vancomycin resistance, 11 (7.3%) of which had an MIC over 8 g/mL [21]. It has been historically proven difficult to treat serious infections due to vancomycin-resistant Enterococci (VRE), require combination therapy and management of treatment-related toxicity. Even though new antibiotics with VRE activity have been introduced to the therapeutic arsenal, significant challenges remain. It is easier for clinicians to tackle these challenging hospital-associated pathogens if they understand the factors driving the emergence of resistance to VRE, the dynamics of gastrointestinal colonization, microbiota-mediated colonization resistance, and mechanisms of resistance to currently available therapeutic options. Daptomycin and linezolid antibiotics inhibit VRE; however, understanding their clinical role and mechanisms of resistance is critical to maximizing their effectiveness [22].

In a meta-analysis performed in Iran among culture-positive Enterococcus species cases, VRE infections were found to be 9.4%. In Germany, the United Kingdom (UK), and Italy, VRE prevalence was 11.2%, 8.5–12.5%, and 9%, respectively. Moreover, the rate of vancomycin resistance among *E. faecalis* isolates was higher than for *E. faecium* [23]. In the present study there were 6.47% cases of vancomycin-resistant Enterococcus. The prevalence of VRE was more in *E.faecalis* (5.86%) compared to *E.faecium* (3.51%). In another meta-analysis performed in Ethiopia, the analysis included 831 Enterococci and 71 VRE isolates. VRE prevalence was 14.8%. Enterococci were more resistant to penicillin (60.7%), amoxicillin (56.5%), doxycycline (55.1%), and tetracycline (53.7%) than vancomycin. Daptomycin and linezolid showed relatively low resistance rates with a pooled estimate of 3.2%. Multidrug resistance (MDR) was 60.0% for enterococci [24]. In the present study, there were 23.18% isolates susceptible to ampicillin, 26.14% susceptible to ciprofloxacin, and 26.68% susceptible to levofloxacin. In this study, it was found that 67.92% isolates were multi-drug resistant. The emerging problem of multidrug resistance poses a problem for clinicians as there are fewer therapeutic options. 

Different methods have been documented for detecting linezolid resistance in Enterococcus strains. The Vitek 2 system showed poor correlation between MICs in the susceptible and intermediate range and G2576T mutation status, likely reflecting the lack of validation of the Vitek AST GP-61 card for LR Enterococcus strains. The use of disk diffusion testing appears to be less sensitive than dilution methods for detecting reduced linezolid susceptibility due to the G2576T mutation. However, it is more specific for detecting fully susceptible strains. Variability in E-test results may be due to the difficulty in interpreting 80% growth inhibition end points. Agar and broth dilution methods were in agreement with polymerase chain reaction detection of the mutation, and disk diffusion was somewhat less sensitive, but equally specific. Although the first report of a strain of linezolid-resistant enterococcus was reported from Kolkata [6], we detected two strains that were linezolid-resistant. Both isolates were found to have found to have G2576T mutation in 23S rRNA. According to a study in Katihar, 2.8% of enterococcus isolates were resistant to linezolid [25]. In the current study, 99.46% isolates were susceptible to linezolid. However, the development of linezolid resistance is an alarming feature as linezolid is one of the last resorts for management of VRE. Hence, screening for linezolid resistance and understanding the mechanism of linezolid resistance are essential for proper management of infections caused by linezolid-resistant Enterococcus species.

By both disk diffusion and agar-screen methods, greatest resistance to aminoglycoside was observed among *E. faecium*, followed by *E. durans*, *E. faecalis*, and *E. casseliflavus*. The high-level aminoglycoside resistance (HLAR) was significantly (*p* < 0.05) more prevalent in *E. faecium* [26]. Two-hundred and fifteen (57.95%) of the Enterococcus in this study were susceptible to high-level gentamicin. However, there was no significant difference in the prevalence of susceptibility to high-level gentamicin among the different species.

According to a study performed in China, among 1157 clinical isolates of Enterococcus species, there were 679 E. *faecium* isolates (58.7%), 382 *E. faecalis* isolates (33%), 26 *E. casseliflavus* isolates (2.2%), 24 *E. avium* isolates (2.1%), and 46 isolates of other Enterococcus species (4%). Significantly different prevalence of antimicrobial resistance between *Enterococcus faecium* and *Enterococcus faecalis* was observed, and ≤ 1.1% of these Enterococcus species were resistant to vancomycin, teicoplanin, or linezolid. Moreover, different Enterococcus species isolated from different departments of the hospital exhibited different levels of resistance to the same antimicrobial agent, while reserpine treatment significantly reduced resistance to ciprofloxacin, gatifloxacin, and levofloxacin [27].

The susceptibility rates of vancomycin-resistant *E. faecium* urinary isolates were 100% for linezolid, 81% for fosfomycin, 68% for tetracycline, 6% for ampicillin, and 3% for penicillin [28]. In a study performed to assess fosfomycin susceptibility among the VRE isolates, it was found that 26.6% of bacteria were susceptible to fosfomycin [29]. Researchers found that fosfomycin combined with daptomycin or daptomycin monotherapy had bactericidal effects against VRE at 24 h in an in vitro study [30]. There was a study performed in China in which 19 VRE isolates were resistant to fosfomycin, 18 of which had conjugative fosfomycin resistance and were positive for the fosB gene [31]. In the present study, 97.76% of the Enterococcus isolates were susceptible to fosfomycin. However, genotypic characterization of the fosfomycin resistance was beyond the scope of this study. 

In a study performed in India among 514 isolates of Enterococci with 46.5% *Enterococcus faecalis* and 53.5% *Enterococcus faecium, E. faecalis* was seen to be significantly more resistant (*p* < 0.05) to ciprofloxacin, and high strength gentamicin. 7.2% isolates were resistant to vancomycin. Among these, 114 (22.18%) isolates were MDR [32]. One study, conducted in Iran, found high-level gentamicin resistance in 50.9% of isolates, though all isolates were multidrug-resistant (100%) [33]. According to another study, 93% of isolates studied were resistant to one or more antimicrobial agents, including tetracycline (86%), ciprofloxacin (73%), and quinupristin-dalfopristin (53%). High-level gentamicin and high-level streptomycin resistance were detected in 50% and 34% of isolates, respectively [34]. In the current study, 57.95% of the Enterococcus were susceptible to high-level gentamicin. 

The limitations of the study were that genotypic characterization of the fosfomycin resistance was beyond the scope of this study. Moreover, follow-up of the patients was not performed to look for the clinical improvement of the infection. 

## 5. Conclusions

This study showed a prevalence of 24 (6.47%) vancomycin-resistant Enterococcus, 2 (0.54%) linezolid-resistant Enterococcus, and 252 (67.92%) multi-drug-resistant Enterococcus. In the current era of developing resistance, it is essential to characterize the different resistant isolates for proper management of these cases. The intrinsic resistance of different Enterococcus species to vancomycin should also be noted. 

## Figures and Tables

**Table 1 diagnostics-13-00945-t001:** Sampling location of 371 *Enterococcus* spp. isolates obtained in this study.

Sample	Number (%)
Urine	223 (60.11%)
Blood	76 (20.48%)
Pus	68 (18.33%)
Tissue	4 (1.08%)

**Table 2 diagnostics-13-00945-t002:** The Enterococcus species isolated along with Vancomycin resistance detected.

Enterococcus Species	Number (%)	No of VRE	Gene Detected
*Enterococcus faecalis*	239 (64.42%)	14	VanA
*Enterococcus faecium*	114 (30.72%)	4	VanA
*Enterococcus avium*	8 (2.16%)	-	-
*Enterococcus durans*	4 (1.08%)	-	-
*Enterococcus casseliflavus*	4 (1.08%)	4	VanC
*Enterococcus gallinarum*	2 (0.54%)	2	VanC

**Table 3 diagnostics-13-00945-t003:** Antimicrobial susceptibility of *Enterococcus* spp. (*n* = 371).

Antimicrobial Agent	*Enterococcus* spp. (*n* = 371)	MIC Range μg/mL	MIC_90_ μg/mL	MIC_50_ μg/mL
Ampicillin/Penicillin	86 (23.18%)	≤2–≥32	32	32
Ciprofloxacin	97 (26.14%)	≤0.5–≥8	8	8
Levofloxacin	99 (26.68%)	≤0.5–≥8	8	8
High-level gentamicin	215 (57.95%)	-	-	-
Vancomycin	347 (93.53%)	≤0.5–≥32	1	0.5
Teicoplanin	353 (95.15%)	≤0.5–≥32	1	0.5
Linezolid	371 (99.46%)	≤0.5–4	1	0.5
Tetracycline	137 (36.93%)	≤0.5–≥16	16	16
Fosfomycin (*n* = 223)	218 (97.76%)	≤0.5–512	8	2
Nitrofurantoin (*n* = 223)	192 (86.1%)	≤16–256	128	16
Erythromycin (*n* = 148)	62 (41.89%)	≤0.5–≥8	8	0.5
Chloramphenicol (*n* = 148)	76 (51.35%)	-	-	-

**Table 4 diagnostics-13-00945-t004:** Relation of antimicrobial susceptibility among *E. faecalis* and *E. faecium*.

Antimicrobial Agent	*E. faecalis* (*n* = 239) (No Indicates Susceptible Isolates)	*E. faecium* (*n* = 114) (No Indicates Susceptible Isolates)	*p* Value
Ampicillin	72	14	0.0002
Ciprofloxacin	65	32	0.898
Levofloxacin	69	30	0.704
High-level gentamicin	142	73	0.417
Nitrofurantoin	151	41	0.0001
Tetracycline	98	39	0.243
Erythromycin	41	21	0.766

## Data Availability

Data has been presented in the results section.

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
