# Peer review of "Vancomycin and Linezolid-Resistant Enterococcus Isolates from a Tertiary Care Center in India"

_diagnostics, 2023, doi:10.3390/diagnostics13050945_

Round 1
Reviewer 1 Report
-The current study is interesting; however, the authors should address the following comments to improve the quality of the manuscript:
1. The abstract must illustrate the used methods and the most prevalent results (give more hints about methods and results). Besides, rephrase the aim of the work and the main conclusion of your findings.
2. Discuss more on significant results of this study, specially about linezolid resistance.
3. Please discuss more alarming results of this research in discussion section.
4. Minor spacing problems exist between the words in some parts of the text.
5. The English of the manuscript should be reviewed and syntax and errors should be corrected before publication.
6. It is better to use the latest version of Clinical and Laboratory Standards Institute.
Reviewer 2 Report
This manuscript described the characteristics of enterococci isolated from a tertiary care center in India. The vancomycin resistant isolates were explained with simple Van gene detection. There should be explanation for each table but only one (Table 3) was mentioned in the manuscript and others were not explained well. Please check the names of bacteria and capital letters carefully again. Please see the detailed comments below.
https://www.mdpi.com/journal/diagnostics/instructions
Please check the affiliation names and location again. Affiliation information usually does not include position of the authors. Also, please add the country name ‘India’.
Abstract
Abs_Results_L1: Bacterial genus and species names (Enterococcus) should be italicized. Please check it throughout the manuscript. (Staphylococcus aureus, Enterococcus faecalis)
Introduction
VRE is alarming because of the high mortality by the reference (2) but it doesn’t mean that the high mortality rate made VRE on the rise. Please re-write the sentence.
Second paragraph: How can you treat resistant enterococci with glycopeptide? It should be explained in detail with proper references or re-write the sentence.
Please add ‘, India’ after the location name Assam since it’s an international journal and it would be better for readers to understand.
Materials and methods
M&M_L3: Abbreviations (CSF) should be explained at its first appearance.
What are the standard microbiological techniques? There should be at least one reference for the methods (identification and AMR test) used in this study.
Data analysis – excel -> Excel. Please add a product information for SPSS. Chi square -> Chi-square
Although this journal accepts free-format submission, Tables should have a short explanatory title and caption. Please change the titles for the tables. For example, Table 1. Sampling location of 371 Enterococcus spp. isolates obtained in this study.
Results: Many MDR isolates were detected and what about their resistance patterns? The resistance patterns of the isolates, such as up to how many classes of antimicrobials were resistance, should be explained.
Do the numbers in the Table 4 indicate susceptible isolates or resistant isolates?
Round 2
Reviewer 1 Report
After the changes made, the manuscript can be published.
Author Response
As the reviewer has not suggested any further changes, Kindly accept the article for publication.
Reviewer 2 Report
The authors should address point-to-point response to the review comments.
Comment such as 'modified as suggested' cannot give full explanation for the comments and It is hard to find the revised writing for some of the writing.
Introduction: VRE is alarming because of the high mortality by the reference (2) but it doesn’t mean that the high mortality rate made VRE on the rise. Please re-write the sentence.
Introduction Second paragraph: How can you treat resistant enterococci with glycopeptide? It should be explained in detail with proper references or re-write the sentence.
Results: Many MDR isolates were detected and what about their resistance patterns? The resistance patterns of the isolates, such as up to how many classes of antimicrobials were resistance, should be explained.
The resistance rate can be found from the Table 3 but it is hard to find up to how many classes of antimicrobials from each isolate found to be resistant. To see the resistance patterns of the MDR isolates, please provide the overall resistance of isolates in this study.
Author Response
Please find the attached explanation
All changes are made in track changes mode in red

Round 3
Reviewer 2 Report
The manuscript was revised as suggested and the point-to-point response was provided by authors.